# Use of Ultra-Widefield Fluorescein Angiography to Guide the Treatment to Idiopathic Retinal Vasculitis, Aneurysms, and Neuroretinitis—Case Report and Literature Review

**DOI:** 10.3390/medicina58101467

**Published:** 2022-10-16

**Authors:** Ping-Ping Meng, Chun-Ju Lin, Ning-Yi Hsia, Chun-Ting Lai, Henry Bair, Jane-Ming Lin, Wen-Lu Chen, Yi-Yu Tsai

**Affiliations:** 1Department of Ophthalmology, China Medical University Hospital, China Medical University, Taichung City 404, Taiwan; 2School of Medicine, College of Medicine, China Medical University, Taichung City 404, Taiwan; 3Department of Optometry, Asia University, Taichung City 404, Taiwan; 4Byers Eye Institute, Stanford University School of Medicine, Stanford, CA 94303, USA

**Keywords:** ultra-widefield fluorescein angiography, idiopathic retinal vasculitis, aneurysms and neuroretinitis, panretinal laser photocoagulation, case report

## Abstract

Purpose: To review the clinical features, diagnosis, and treatment of idiopathic retinal vasculitis, aneurysms, and neuroretinitis (IRVAN) and to report a case with the use of ultra-widefield fluorescein angiography (UWFA) for confirming the precise staging of IRVAN and aid in early treatment. The patient improved after being treated with intravitreal aflibercept injection. Results: A 26-year-old female complained of progressive blurred vision OD for one week. Her BCVA was 0.6 OD and 1.0 OS. Fundus examination showed vitritis, retinal hemorrhage, and vasculitis over bilateral eyes. Fluorescein angiography (FA) with a 55 degree of view revealed aneurysmal dilations of the peripapillary arteriole, peripapillary focal leakage, venous leakage, and capillary nonperfusion area. Stage 2 IRVAN was impressed OU. Oral prednisolone was administered. After four months, she experienced decreased visual acuity OS. Optical coherence tomography (OCT) revealed subretinal and intraretinal fluid with hyperreflective material. One posterior subtenon triamcinolone and one intravitreal aflibercept injection were performed OS, and macular edema subsided. A 105-degree ultra-widefield fluorescein angiography (UWFA) showed multiple peripheral background hypofluorescence areas corresponding to capillary nonperfusion. Retinal neovascularization (NV) was found OS, which had not been revealed by the previous 55-degree FA. Stage 3 IRVAN was made OS and panretinal laser photocoagulation (PRP) was performed. Oral prednisone and cyclosporine were prescribed. Her vision improved to 1.0 OU. Conclusion: UWFA provides visualization of peripheral retinal pathology and for precise staging. It also had direct implications in the follow-up and treatment strategy.

## 1. Introduction

IRVAN is characterized by features of vasculitis, multiple aneurysmal dilatation of retinal arterioles and optic nerve arterioles, and neuroretinitis. IRVAN is rare and mostly occurs in young, healthy female individuals [1]. Samuel et al. published a case series with 22 patients and divided the clinical entity into five stages [2]. They imply that early diagnosis and treatment are important and that the visual prognosis is dependent on early panretinal laser photocoagulation over the nonperfusion area, with best visual outcomes in stage 2 (angiographic evidence of capillary nonperfusion), before or shortly after the development of neovascularization [2]. Some studies suggest performing laser when there are more than two quadrates of retinal nonperfusion area noted or when there is high-risk proliferative retinopathy with neovascularization (NV) [3]. Fluorescein angiography is useful in the diagnosis of IRVAN and may reveal ophthalmoscopically invisible vascular changes and the circulatory dynamics of retinal vasculature. The importance of recognizing nonperfusion areas highlight the importance of ultra-widefield fluorescein angiography in detecting these lesions.

## 2. Case Presentation

A 26-year-old female complained of progressive blurred vision in the right eye for one week. Her past medical history included Bell’s palsy 10 years ago and psychotic disorder. She had a history of contact with a cat. Family history included a mother with rheumatic arthritis and Sjogren’s syndrome and a father who was a tuberculosis suspect.

Her BCVA was 0.6 in the right eye and 1.0 in the left eye. No fever, no headache, no lymphadenopathy, no oral ulcer, no diarrhea, no neck stiffness, no lumbar pain, no arthralgia, no urethralgia, no skin rash nor open wound was mentioned. No abnormal findings were present in the anterior segment, but fundus examination showed bilateral vitritis, retinal hemorrhage, and vasculitis (Figure 1). FA with a 55-degree view revealed aneurysmal dilations of peripapillary arteriole, peripapillary focal leakage, venous leakage, and capillary nonperfusion area (Figure 2). Autofluorescence images highlighted aneurysmal dilations of retinal arterioles and vein tortuosity (Figure 3). OCT showed focal disruption in both the ellipsoid and retinal pigment epithelium (RPE)–photoreceptor interdigitation zones over the right eye (Figure 4).

Our initail diagnosis is retinal vasculitis over both eyes. We discussed with a rheumatologist the suspicion of systemic vasculitis, and series test was arranged, including complete blood count with differential count (CBC/DC), hsCRP, antineutrophil cytoplasmic antibodies, rheumatic factor, complement 3, and complement 4, which were all negative. The patient denied any of the other systemic symptoms. Only mild elevated erythrocyte sedimentation rate (ESR) was found. The following systemic diseases were ruled out by the corresponding negative tests results: Sjogren’s syndrome (negative Anti-ENA II (SS-A and SS-B)); Antiphospholipid syndrome (Anti-Cardiolipin IgG); Typhus and spotted fever (Weil-Felix test). Toxoplasmosis IgG and IgM and Quantiferon tests were negative as well. In addition, the results for anterior chamber fluid polymerase chain reaction tests for cytomegalovirus, herpes simplex virus, Epstein-Barr virus, and varicella-zoster virus were all negative. Additionally, during the follow up period of three years, no systemic symptoms nor any neurologic signs were revealed. Therefore, we excluded the majority of possible systemic inflammatory and infectious etiology.

A clinical diagnosis of stage 2 IRVAN in both eyes was made due to all three major criteria being met, including retinal vasculitis, aneurysmal dilations of retinal arterioles, and neuroretinitis, and stage 2 was confirmed with the evidence of capillary nonperfusion.

Oral 10 mg prednisolone daily was administered, but ocular signs and symptoms still persisted. After four months, she experienced decreased visual acuity in the left eye. OCT revealed subretinal and intraretinal fluid with hyperreflective material in the left eye (Figure 5). One posterior subtenon triamcinolone and one intravitreal aflibercept injection were performed in the left eye and macular edema subsided. The Heidelberg Spectralis UWFA imaging system (Heidelberg Engineering, Heidelberg, Germany) provides a 105-degree peripheral retinal image, showing multiple peripheral background hypofluorescence areas corresponding to capillary nonperfusion. Retinal neovascularization was found in the left eye, which had not been revealed by the previous 55-degree FA (Figure 6). A diagnosis of stage 3 IRVAN was made in the left eye, and we performed panretinal laser photocoagulation on the nonperfusion area to decrease the risk of visual loss. Oral prednisone 15 mg and cyclosporine 100 mg daily were also prescribed. The patient’s visual acuity improved to 1.0 in both eyes after the treatment.

## 3. Discussion

We report a case of IRVAN using ultra-widefield fluorescein angiography to detect peripheral retinal vasculitis, non-perfusion area, and NV, aiding in earlier diagnosis and treatment. In addition, due to the ineffectiveness of the initial steroid treatment, we used intravitreal aflibercept injection for further management, which was successful in relieving the macular edema.

IRVAN is on the differential diagnoses of retinal vasculitis and its name highlights its salient features, including multiple aneurysmal dilatations of retinal arteriole, optic nerve head arterioles, the retinal vasculitis and neuro-retinitis, the leakage from vessels, and staining of the optic nerve can be observed on FA [2,4]. In IRVAN, retinal arteritis is more prominent, unlike most vasculitis involving mainly phlebitis, and the aneurysm, which may be Y-shaped or fusiform, is present at or near the branching point of retinal arterioles [1]. The diagnosis of ocular diseases with retinal vasculitis can be differentiated based on the involving vessels. When arteritis predominates, the condition could be caused by systemic lupus erythematosus, polyarteritis nodosa, syphilis, herpes simplex virus infection, varicella-zoster virus infection or IRVAN. When phlebitis predominates, it could be sarcoidosis, multiple sclerosis, Bechet disease, birdshot chorioretinopathy, HIV infection, tuberculosis or Eales disease. If the vasculitis involves both vein and artery, toxoplasmosis, granulomatosis with polyangiitis, Crohn’s disease, frosted branch angiitis could be the cause [5,6].

The pathophysiology of IRVAN is not well defined but inflammation is integral to the disease [1]. The artery anterior to the equator contains only one to two layers of muscular wall, as does the retinal vein. Inflammation primarily involves the smooth muscle layer of the arterial wall and leads to focal loss and weakening of the arterial wall and later aneurysmal dilatation. Sometimes, intimal desquamation and proliferation leads to leakage, causing macular edema. This explains the predilection of an involvement vessel that is posterior to the equator [1]. The pathogenesis of neuroretinitis is due to the direct involvement of the viral, bacterial, or other atypical infection or inflammation of the optic nerve, causing peripapillary swelling and exudation. The deposition may extend to the macula, leaving a macular star appearance [7]. However, in the condition of IRVAN, it is actually the exudative maculopathy that masquerades as neuroretinitis [7].

The consistent feature of IRVAN is a peripheral nonperfusion area, owing to increased vessel permeability and persistent macular edema, and with lipid deposition, causes the chronic upregulation of vascular endothelial growth factor (VEGF) from the ischemic peripheral retina [1]. Therefore, there is a role for anti-VEGF treatment and PRP for retinal ischemia or NV, which can preserve vision and prevent disease progression [6]. There is no consensus on the treatment of IRVAN. Treatment modalities include observation, photocoagulation, cryotherapy, intraocular or oral steroid, intravitreal bevacizumab or ranibizumab, pars plana vitrectomy, and infliximab, depending on stages and complications of the disease [8]. Inflammation predominates during the early stages; therefore, steroids and immunosuppressant drug are most effective during this period. In later stages, treatments targeting ischemia and NV were used [9,10]. Samuel et al. reported that early PRP treatment may be considered in areas of widespread nonperfusion in the retina, before NV development [9]. Rouvas et al. suggested that PRP treatment should be withheld until the ischemic area involvement is in more than two quadrants and is better observed by wide-field FA [11]. Additionally, it is contraindicated to perform laser over a leaking arterial aneurysm because it may lead to blood vessel occlusion [3]. FA, usually with a 55-degree of view, is essential to the diagnosis of IRVAN but is limited in its ability to detect peripheral capillary nonperfusion areas and NV. Sweeping the image is possible with the ultra-widefield 105-degree lens to allow greater coverage of the retinal periphery. Ultra-widefield FA is a useful tool for detecting peripheral retinal ischemia, which may have direct implications in the diagnosis, follow-up, precise staging, and treatment, such as targeted peripheral photocoagulation. Few studies have concluded that the mangement of noninfectious uveitis changed based on the results of UWFA [12,13]. We report a case of IRVAN in Taiwan, which was complicated with NV and successfully treated with intravitreal aflibercept, photocoagulation, steroid, and immunomodulatory agents.

## 4. Conclusions

Ultra-widefield FA provides the visualization of pathology of retina periphery, which could not be obtained from traditional FA. In IRVAN, it could be utilized for detecting peripheral retinal NV and nonperfusion area and for early treatment due to precise staging. The use of ultra-widefield FA also had direct implications in the follow-up and treatment strategy.

## Figures and Tables

**Figure 1 medicina-58-01467-f001:**
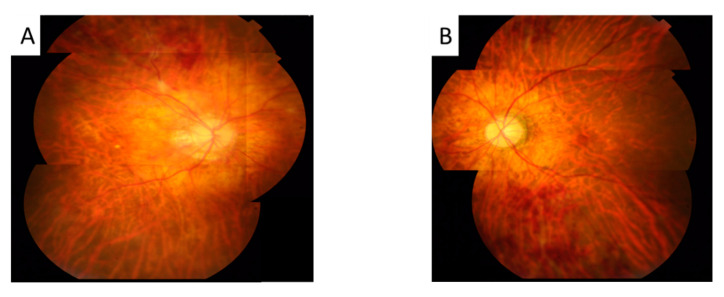
The color photos showed mid vitritis, retinal exudates, and hemorrhage in right eye (**A**) and left eye (**B**) both eyes.

**Figure 2 medicina-58-01467-f002:**
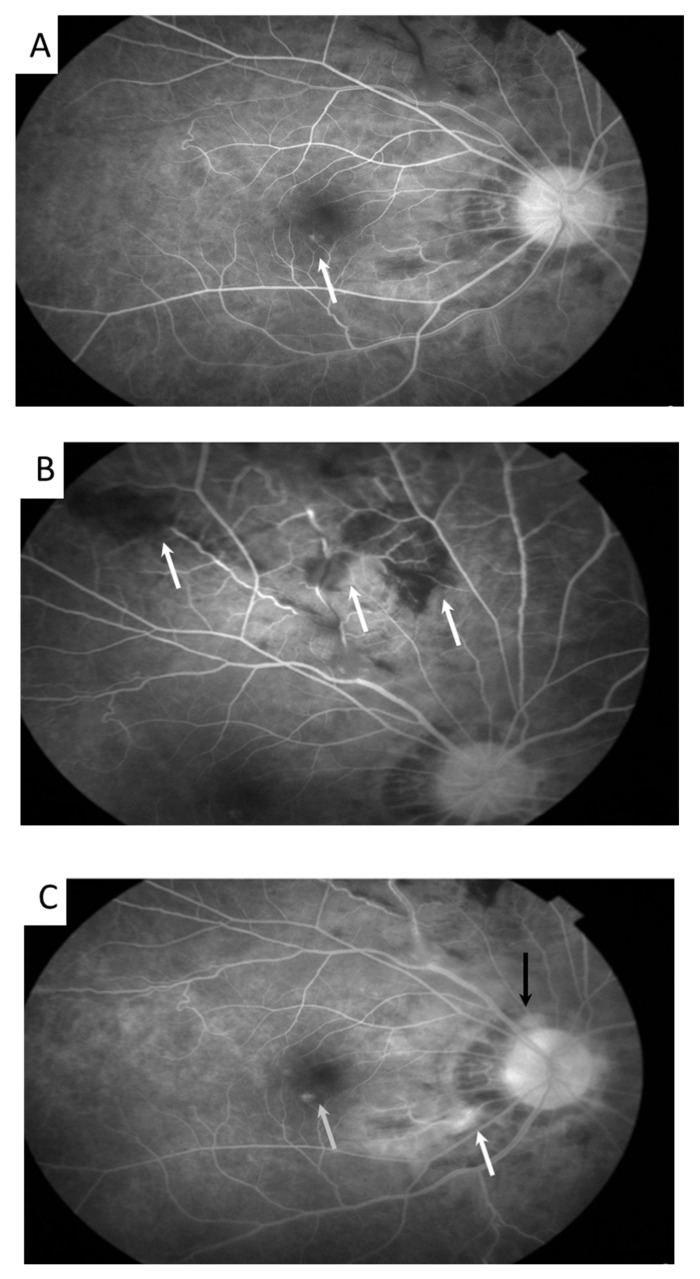
Traditional fluorescein angiography showed aneurysmal dilation ((**A**), 40 s after the fluorescein injection, arrow) of peripapillary and optic disc retinal arterioles, peripheral capillary nonperfusion ((**B**), 1 min after the fluorescein injection, arrow), focal inflammation of the optic nerve ((**C**), 4 min after fluorescein injection, black arrow), peripapillary retina ((**C**), white arrow), macula ((**C**), gray arrow), and retinal vasculitis in the right eye.

**Figure 3 medicina-58-01467-f003:**
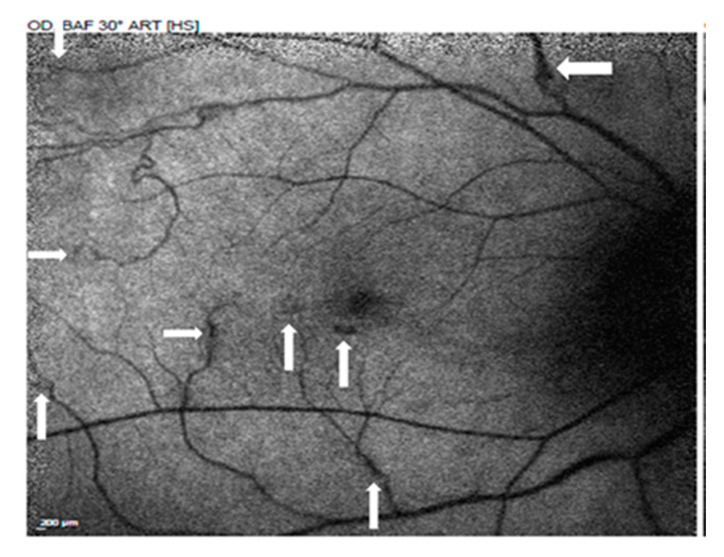
Autofluorescence images highlighted the aneurysmal dilations (arrow) of retinal arterioles and vein tortuosity.

**Figure 4 medicina-58-01467-f004:**
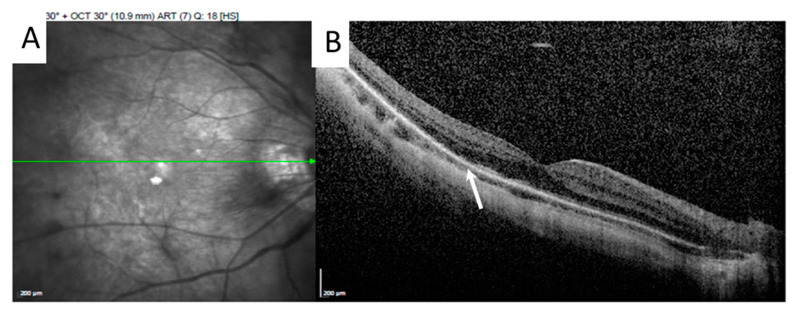
The green line over the infrared red image (**A**) represents the OCT scan position. High-resolution OCT showed focal disruption of both the ellipsoid and RPE–photoreceptor interdigitation zones ((**B**), arrow) over the right eye.

**Figure 5 medicina-58-01467-f005:**
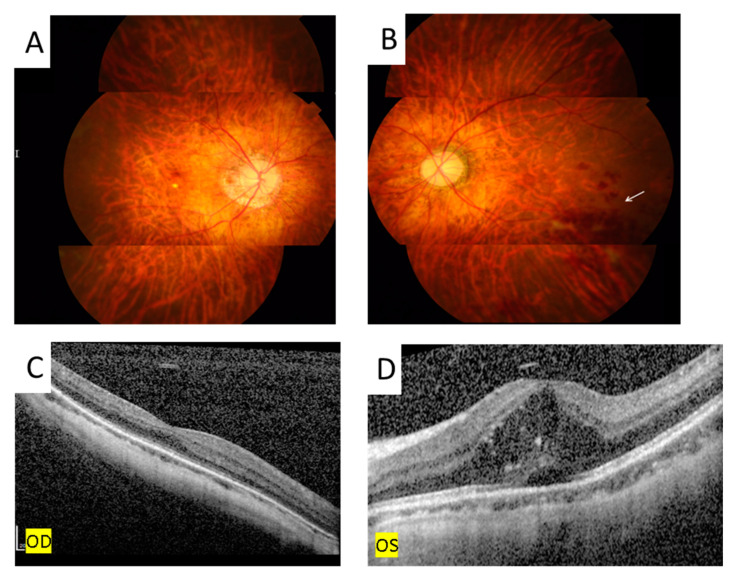
Fundus photography showed retinal hemorrhage ((**B**), arrow) and OCT disclosed subretinal and intraretinal fluid with hyperreflective material over the left eye (**D**). There is no retinal hemorrhage (**A**) nor macular edema (**C**) over the right eye.

**Figure 6 medicina-58-01467-f006:**
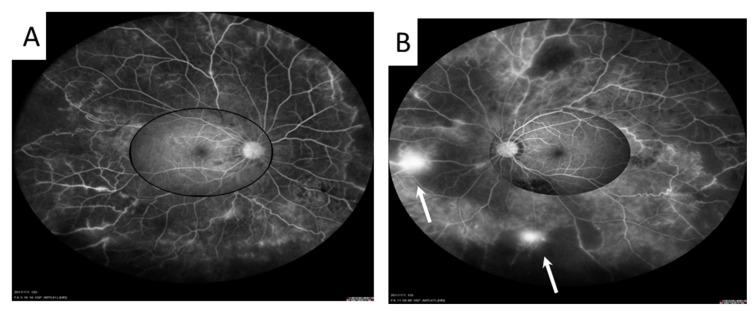
The Heidelberg Spectralis UWFA imaging system (Heidelberg Engineering, Heidelberg, Germany) provides a 105-degree, high-contrast, peripheral retinal image. It showed widespread peripheral vasculitis and capillary nonperfusion in both eyes (**A**,**B**). Retinal neovascularization ((**B**), arrow) in the left eye was found, which was not revealed by the previous traditional FA.

## Data Availability

All the data supporting our findings will be shared upon request, although the majority is contained within the manuscript.

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
