# Peer review of "Use of Ultra-Widefield Fluorescein Angiography to Guide the Treatment to Idiopathic Retinal Vasculitis, Aneurysms, and Neuroretinitis—Case Report and Literature Review"

_medicina, 2022, doi:10.3390/medicina58101467_

Round 1

Reviewer 1 Report (Previous Reviewer 2)

The manuscript is highly improved.

Unfortunately, the possibility of bartonellosis (with fever, lymphadenopathy, scrab or pustule) and sclerosis multiplex were not excluded.

Please be included: the patient did not mention any symptoms for prior cat scratch disease and during the follow-up period (how long was it?) no neurological/ psychiatric signs were observed

I favor the publication of this manuscript but an MRI of the brain should be done just for the interest of the patient.

Author Response

Thanks so much for the suggestions and questions.

Indeed, bartonellosis and multiple sclerosis are of the differential diagnosis of the patient’s condition. I’m sorry that we didn’t provide sufficient negative findings of the systemic condition and we have added it in the following paragraph. (the revised part had been marked RED)

Please see Case presentation section, paragraph 2, line 1-3

No fever, no headache, no lymphadenopathy, no oral ulcer, no diarrhea, no neck stiffness, no lumbar pain, no arthralgia, no urethralgia, no skin rash nor open wound was mentioned.

Please see Case presentation section paragraph , line 12-13  

Also, during the follow up period of three years, no systemic symptoms nor any neurologic symptoms was revleaed.

There is no neurologic symptoms and signs of this patient for the follow up period of three years and I will track the brain image once the patient return to the clinic in the future.  

Reviewer 2 Report (Previous Reviewer 3)

The authors have addressed the issues raised in the review in a sufficient manner.

Author Response

Thanks so much for the suggestions and for reviewing our paper.

This manuscript is a resubmission of an earlier submission. The following is a list of the peer review reports and author responses from that submission.

Round 1

Reviewer 1 Report

The article presents a case of retinal vasculitis, presumed to be IRVAN. However, the data presented (AFG especially) does not fully support this diagnosys. There is no evident sign of neuroretinitis, nor arterial anevrisms.

Wide field AFG makes periferal dettection easier. However, periferal scanning is possible with conventional AFG. I consider that periferal lessions would have been easily detected with conventional AFG and photocoagulation indicated earlier, before macular exudation.

Differential diagnosis is lacking. One of the main diseases to consider (phlebitis, peripheral nonperfusion etc.) should be Eales.

Author Response

The authors thank the editors and the reviewers for the comprehensive review and valuable comments on our manuscript. These opinions are very instructive and precious. We express our uppermost gratitude to you. We have revised the paper in accordance with the reviewers’ suggestions. The questions and opinions raised by the reviewers are answered in sequence below in a point-by-point fashion.

Our reply to the reviewer's question is in the attached file. 

Reviewer 2 Report

Nicely documented manuscript. The figures fulfil strong evidence that ultra-wildfield fluorescein angiography is a useful tool for detection of peripheral vascular complications.

The manuscript is regrettably not persuaded by the facts of evaluation. Case history: IRVAN is the second “idiopathic” inflammatory event in the patient’s life. The first event could be the Bell’s palsy (term for “idiopathic” facial paresis). Were they really “idiopathic”? Please describe in details the steps of differential diagnosis: how was the infective/ allergic- immunologic origin excluded? were diagnostic vitrectomy, brain MR imaging considered?

The value of manuscript could highly be improved by construction of a table about the problems of differential diagnosis based on the literature data

Author Response

The authors thank the editors and the reviewers for the comprehensive review and valuable comments on our manuscript. These opinions are very instructive and precious. We express our uppermost gratitude to you. We have revised the paper in accordance with the reviewers’ suggestions. The questions and opinions raised by the reviewers are answered in sequence below in a point-by-point fashion.

Please see the attachment to see our reply of the reviewer's question. 

Reviewer 3 Report

The manuscript entitled "Use of Ultra-Widefield Fluorescein Angiography to Guide the Treatment to Idiopathic Retinal Vasculitis, Aneurysms, and Neuroretinitis - Case Report and Literature Review" is based on an interesting case report of a patient with idiopathic retinal vasculitis, aneurysms, and neuroretinitis diagnosed with ultra-widefield fluorescein angiography and clinical. This study provides objective measurements and suggestions for diagnosing this rare condition. 

The topic adds to the current literature and is of clinical interest, especially considering the rarity of the disease, and difficulty in diagnosis. Regular fluorescein angiography missed the diagnosis, thus ultra-widefield proved to be essential. The paper is well presented and can be of clinical use.

The only suggestion that could make the paper of practical use is to add a simple flow-chart diagram showing how to handle patients like listing clinical and instrumental testing that can help in the differential diagnosis.

Author Response

The authors thank the editors and the reviewers for the comprehensive review and valuable comments on our manuscript. These opinions are very instructive and precious. We express our uppermost gratitude to you. We have revised the paper in accordance with the reviewers’ suggestions. The questions and opinions raised by the reviewers are answered in sequence below in a point-by-point fashion.

Please see the attachment to see our reply to the reviewer's question. 

Round 2

Reviewer 2 Report

The manuscript v2 is highly improved.